# Quantitative PCR for the Diagnosis of HCMV Pneumonia in HSCT Recipients and Other Immunocompromised Hosts

Carla Berengua [1,2,3,*] and Rodrigo Martino [3,4]

1  Genetics and Microbiology Department, Universitat Autònoma de Barcelona, 08193 Barcelona, Spain
2  Microbiology Department, Hospital de la Santa Creu i Sant Pau, 08041 Barcelona, Spain
3  Sant Pau Institute of Biomedical Research (IIb Sant Pau) Barcelona, 08041 Barcelona, Spain
4  Hematology Department, Hospital de la Santa Creu I Sant Pau, 08041 Barcelona, Spain
*  Correspondence: cberengua@santpau.cat

**Abstract:** Pneumonia is among the most serious manifestations of HCMV infection, with high morbidity and mortality. Probable pneumonia is defined as the detection of HCMV in bronchoalveolar lavage (BAL) by viral isolation or DNA quantification (qPCR) combined with symptoms and/or signs of respiratory infection. However, currently, there is no reproducible and well-defined viral load (VL) from BAL that can reliably differentiate patients with pneumonia from the much more common detection of viral DNA in seropositive patients without true HCMV pneumonia. Several studies have been published with the aim of establishing an optimal VL for differentiating pneumonia from viral lung shedding. The aim of this review is to collect and analyze the methodology and the conclusions obtained in studies whose objectives included the correlation between HCMV VL in BAL and/or the plasma and the occurrence of HCMV pneumonia. For this purpose, a total of 14 articles have been included. There are some conclusions on which they all agree. PCR techniques were more sensitive and had a higher NPV than culture techniques but were less specific and had a low PPV. The mean HCMV loads in both BAL and the plasma were significantly higher in patients with pneumonitis than in those without. The HCMV load in patients with pneumonitis was higher in BAL than in the plasma, making qPCR in BAL a better predictor of HCMV pneumonitis than in the plasma. Nevertheless, this review highlights the difficulty of establishing a universal VL value, both in BAL and in the blood, to differentiate patients with HCMV pneumonia from those without. To complete the information available in these studies, prospective multicentre studies would be required. Methodologically, a large number of patients with HCMV pneumonitis would have to be included, and a subclassification of the type of immunosuppression of each patient should be made in order to obtain an optimal VL threshold in different host groups.

**Keywords:** HCMV; Cytomagalovirus; HSCT; qPCR; bronchoalveolar lavage; viral culture; pneumonia

## 1. Introduction

Human Cytomegalovirus (HCMV) is a double-stranded DNA virus of the family Herpesviridae, subfamily Betaherpesvirinae. It is one of the largest viruses known to cause clinical disease (230 kb). The primary infection usually occurs in the first years of life, by direct contact with secretions of infected individuals, such as saliva, breast milk or urine. The seroprevalence in adults is between 50 and 98%, being higher in developing countries. In the immunocompetent host, the primary infection is usually asymptomatic or presents as a self-limited mononucleosis-like syndrome characterized by fever, lymphadenopathy and lymphocytosis. After primary infection, HCMV remains latent for life in different cells. In immunocompromised hosts (IMC), such as hematopoietic stem cell transplantation (HSCT) or solid organ transplant (SOT) recipients or patients with HIV, uncontrolled viral replication can occur, both after a primary infection, but, most commonly, after reactivation of the latent virus. In these patients, HCMV can cause severe

clinical disease, mainly pneumonitis, colitis or retinitis. The incidence of HCMV disease has decreased in recent years thanks to strategies such as pre-emptive therapy and, more recently, due to prophylaxis in HSCT recipients. Nevertheless, HCMV remains one of the most feared opportunistic pathogens in IMC, especially in those with impaired T-cell mediated immunity [1,2], and is certainly one of the viruses for which HSCT clinicians screen for the most both before and after transplantation.

Pneumonia is among the most serious manifestations of HCMV infection, with high morbidity and mortality, even with antiviral treatment. Allogeneic HSCT recipients are particularly at high risk of HCMV pneumonia, with an extremely variable incidence nowadays (from <1% to 30% of patients with HCMV reactivation) and a mortality of up to 70% [1–4]. The radiological findings and symptoms are nonspecific, which makes diagnosis difficult and delayed. This is paradoxically a more important clinical conundrum nowadays since true HCMV pneumonia has decreased while the development of pneumonias of unknown origin has increased. In chest radiographs, HCMV pneumonitis typically manifests as diffuse interstitial infiltrates, although a nodular pattern may be observed, and completely nonspecific lung infiltrates may also be due to HCMV pneumonia [1,5].

To define and unify the concepts of HCMV infection and disease for use in clinical trials, a consensus report was developed at the Fourth International Conference on HCMV in Paris in 1993. Since then, significant advances have been made in the diagnosis and treatment of HCMV infection, leading to several guideline updates, the latest in 2017 [6]. It differentiates between proven and probable HCMV pneumonia. For proven pneumonia, symptoms and/or signs of respiratory infection are needed with the presence of HCMV identified in lung tissue (i.e., biopsy or autopsy) by virus isolation, rapid culture, histopathology, immunohistochemistry, or DNA hybridization techniques. Probable pneumonia is defined as the detection of HCMV in bronchoalveolar lavage (BAL) by viral isolation or DNA quantification (qPCR) combined with symptoms and/or signs of respiratory infection. However, currently, there is no reproducible and well-defined viral load (VL) from BAL to differentiate patients with pneumonitis from the much more common detection of viral DNA in seropositive patients without true HCMV pneumonia. This problem is, of course, not limited to HCMV but is also the case for other human herpesviruses, especially HHV-6, and Pneumocystis jirovecii, to cite only a few of the BAL "positive PCR" difficult-to-interpret pathogens. This makes it difficult to interpret the results of qualitative reverse-transcriptase PCR. At least in theory, a quantitative PCR (qPCR) would be more helpful in identifying patients at high risk of having true HCMV pneumonia for clinical decision-making. However, qPCR from BAL is far from being standardized.

Definitive pneumonia requires analyzing lung tissue, but obtaining lung biopsies for the detection of HCMV by immunohistochemistry requires invasive techniques, which are nearly always contraindicated, especially in HSCT recipients. Currently, the use of BAL samples for the diagnosis of pneumonia in these patients is already widely extended and accepted [7].

In the last 20 years, the laboratory methods used to detect the virus have also been changing with the development and introduction of molecular techniques. However, one should never forget that proving the presence of a viable, replicating virus is the only way of extrapolating active disease in the clinical sample tested. Thus, viral culture is considered the gold standard for the diagnosis of active HCMV disease, either by traditional tube culture or by culture through centrifugation (shell vial). These allow the recovery of the virus that is replicating and has a high correlation with the occurrence of disease [1,8]. The main limitation of traditional culture is the time needed to observe the characteristic cytopathic effect of HCMV replication, which can range from 1 to 4 weeks, which limits its usefulness in diagnosis and clinical decision-making. However, thanks to the centrifugation of the culture in a shell vial, it is possible to obtain results within 24–48 h, which increases its usefulness in clinical practice, as well as being more sensitive than the traditional tube [1,8]. The detection of the HCMV pp65 antigen expressed on leukocytes during the early period of infection (antigenemia) has also been a widely used technique for the

diagnosis and monitoring of HCMV infection. This technique has many limitations, such as a lack of standardization, laboriousness, and the need for samples with an adequate number of leukocytes in the blood (difficult in transplant patients) [8]. Both culture-based techniques and antigenemia have been displaced in most laboratories by simpler, faster, and more sensitive techniques, such as DNA detection by PCR. It should be kept in mind that HCMV persists latently in cells after primary infection without causing disease, and this non-replicating DNA may be detected by PCR techniques. This makes the detection of HCMV DNA a less specific indicator of active virus replication and disease than culture. The widespread use of PCR techniques and the possible misinterpretation of their results (possible detection of non-replicating DNA) may lead to the over-diagnosis of HCMV disease and, consequently, over-treatment [9,10].

Today, DNA quantification using qPCR is the most widely used PCR method for the diagnosis of HCMV infection or monitoring the patients' VL, especially in peripheral blood samples. Blood screening by qPCR for HCMV DNAemia is recommended at least once a week during the high-risk period after transplantation and to monitor response to pre-emptive antiviral treatment. The major limitation of qPCR is a lack of well-established VL thresholds to guide various clinical applications, a consequence, above all, of the variability between the assays marketed for this determination, despite their calibration to the WHO International Standard. This limitation applies to any sample analyzed, but in the case of HCMV pneumonia, its diagnosis using BAL faces further issues that complicate its interpretation. Thus, in addition to the lack of standardization in obtaining BAL samples, their subsequent processing also introduces variability when it comes to quantifying HCMV DNA. Finally, clinically relevant threshold BAL VL certainly varies depending on the type of the patient [8,11].

Several studies have been published with the aim of establishing the optimal VL of HCMV by qPCR in BAL samples for differentiating patients with pneumonia from those without. The quantification of HCMV in the blood (DNAemia) has also been studied for its potential role in diagnosing patients with pneumonia. In both cases, a universal conclusion has not been reached.

The aim of this review is to collect and analyze the methodology and the conclusions obtained in studies whose objectives included the correlation between HCMV VL in BAL and/or the plasma and the occurrence of HCMV pneumonia.

## 2. Materials and Methods

For this purpose, a search was carried out in the Pubmed database. This search consisted of three components: "cytomegalovirus", "bronchoalveolar lavage", and "viral load". "cytomegalovirus" [Title/Abstract] AND "bronchoalveolar lavage" [Title/Abstract] AND "viral load" [Abstract].

Only articles with full-text availability that were written in English between 1992 and October 2022 were included. This search generated 47 results as of 14 November 2022. After reviewing these articles, we excluded those that dealt with ventilator-associated pneumonia or nosocomial pneumonia and those whose primary objectives did not include the study of HCMV pneumonia. This resulted in 14 studies whose objective(s) included the relationship between HCMV DNA load in BAL and HCMV pneumonia.

## 3. Results

The results of the 14 studies analyzed are shown in detail in Table 1.

**Table 1.** Studies on the role of hCMV PCR from BAL in the diagnosis of hCMV pneumonia in immunocompromised patients, in descending chronological order from the date of publication (studies available as of December 2022).

| Article | Material and Methods | Results and Conclusions | Strengths | Limitations |
|---|---|---|---|---|
| Saksirisampant et al., 2022. [12] | - Overall, 45 BALs and the plasma from 45 **IMS adult** patients with clinical and radiological findings compatible with pneumonia were tested by HCMV qPCR. | - Eleven (24%) patients developed HCMV pneumonitis. All of these patients had detectable plasma HCMV VL (median VL 41,939 UI/mL) and detectable BAL HCMV VL (median VL 379,652 IU/mL)<br>- Thirty-four (76%) patients without HCMV pneumonitis, with median plasma and BAL HCMV VL of 0 IU/mL.<br>- A significant positive correlation was observed between plasma and BAL HCMV VL ($R^2$ = 0.887, $p < 0.001$).<br>- A HCMV VL in BAL was established as an optimal cut-off value to distinguish between HCMV pneumonitis and non-pneumonitis: **831 IU/mL in the plasma and 24,565 IU/mL in BAL**.<br>- Undetectable plasma HCMV excluded HCMV pneumonitis. | - BAL and plasma samples were collected in parallel, which were prospectively qPCR on fresh samples.<br>- A fully automated and highly sensitive qPCR assay was used. | - Small number of patients with HCMV pneumonitis (n = 11).<br>- Viral culture was not used to establish virus replication.<br>- Potential bias in selecting patients for bronchoscopy.<br>- Heterogeneous group of immunocompromised patients included.<br>- Not all patients could be studied by histopathology or cytology. |
| Leuzinger et al., 2020. [9] | - In 1109 BALs from 799 **IMS** patients (median age: 61 years), the presence of HCMV was tested by viral culture, specific immunofluorescence, and qPCR.<br>- Overall, 76 patients with a HCMV-positive load in BAL were also tested by HCMV qPCR in the plasma. | - Median HCMV VL was significantly higher in culture-positive than in culture-negative BAL.<br>- The likelihood for HCMV detection by culture in BAL increased with higher VL (85% for VL >10,000 copies/mL).<br>- BAL HCMV VL of **10,000 copies/mL** was indicative of relevant replication.<br>- HCMV VL cut-off is likely to vary depending on BAL procedure and processing or the assay used for qPCR.<br>- One-third of patients with HCMV-positive BAL had undetectable plasma loads, indicating local HCMV replication in the lung. | - High number of patients and samples studied.<br>- Viral culture performed on all BAL as a significantly relevant indicator of replication.<br>- In a group of patients (112), BALs were prospectively analyzed (in parallel culture and qPCR of fresh samples) to assess the effect of sample freezing on virus DNA degradation. | - Most BALs (997) were tested by qPCR retrospectively from frozen samples.<br>- Heterogeneous group of immunocompromised patients included.<br>- qPCR assay not fully automated (requires prior DNA extraction).<br>- Non-availability of lung biopsy to be able to define more accurately the cases of HCMV pneumonia. |

| Article | Material and Methods | Results and Conclusions | Strengths | Limitations |
|---|---|---|---|---|
| Piñana et al., 2019. [11] | - Overall, 144 BALs and the plasma from 123 allogenic hematopoietic stem cell transplant recipients (**allo-HSCT**) with signs and/or symptoms of pneumonia were tested by HCMV qPCR in 2 different hospitals | - The detection of HCMV DNA in BAL is a very common finding in allo-HSCT, since HCMV DNA was detected in 56 (38.9%) of BALs.<br>- Overall, 60% of the patients had non-proven HCMV pneumonia, having VL >500 IU/mL.<br>- Furthermore, 500 IU/mL is unlikely to be discriminative between pneumonia and pulmonary DNA HCMV shedding.<br>- **HCMV VL cut-off is likely to vary depending** on patients' characteristics, BAL procedure and processing or the assay used for qPCR.<br>- Differences in HCMV VL in BAL provided by qPCR assays used in each centre. | - BALs were included consecutively, and fresh samples were tested (prospective).<br>- A highly sensitive real-time qPCR assay was used. | - Reference techniques, such as viral culture, were not used to establish virus replication.<br>- Two hospitals with different qPCR assays are included.<br>- Retrospective data analysis.<br>- qPCR assay not fully automated (requires prior DNA extraction). |
| Beam et al., 2018. [13] | - Thirty-eight BALs and the plasma from 38 **IMS** patients (the majority were transplant recipients, median age: 55.9 years) were tested by HCMV qPCR. In BAL, VL results were adjusted for the number of cells in each BAL (normalized VL) | - There were 17 (44.7%) patients with HCMV pneumonia (6 proven) and 21 (55.3%) without HCMV pneumonia.<br>- Higher HCMV median VL in BAL was observed in patients with proven pneumonitis (>18,200,000 IU/mL), followed by probable cases (1,305,000 IU/mL) and lower VL in possible cases (32,400 IU/mL).<br>- HCMV VL threshold in BAL of **34,800 IU/mL** would identify patients with pneumonitis. | - Normalization results in VL by cells in BAL. | - Small number of proven pneumonitis (6).<br>- Heterogeneous group of immunocompromised patients included.<br>- Viral culture was not used to establish virus replication.<br>- qPCR performed on BAL samples stored at −80 °C (not fresh).<br>- qPCR assay not fully automated (requires prior DNA extraction). |

**Table 1.** *Cont.*

| Article | Material and Methods | Results and Conclusions | Strengths | Limitations |
|---------|---------------------|------------------------|-----------|-------------|
| Lodding et al., 2017. [14] | - Overall, 972 BALs and the plasma from **141 lung transplant** recipients were tested by HCMV qPCR.<br>- In 859 lung biopsies, HCMV was studied by IHC. | - In 145 (15%) BALs, HCMV was detected by qPCR, of which 34 had pneumonia criteria and 111 without.<br>- BAL HCMV VL was consistently higher in episodes with HCMV pneumonia in lung transplant recipients. Median VL in BAL was 32,940 IU/mL in HCMV pneumonia and 1260 IU/mL without pneumonia ($p < 0.001$).<br>- Optimal cut-off HCMV VL in BAL for diagnosing HMCV pneumonia is **4545 IU/mL** (91% sensitivity and 77% specificity).<br>- qPCR in BAL had high diagnostic accuracy for diagnosing HCMV pneumonia and was better than qPCR in plasma. BAL HCMV VL was log10 1.4-fold higher than the corresponding positive HCMV VL in plasma.<br>- Episodes of HCMV pneumonia were more likely to be HCMV-positive in the plasma (63%) compared with those without pneumonia (24%) ($p < 0.001$).<br>- Overall, 37% of the HCMV pneumonia episodes had a negative PCR in the plasma; thus plasma HCMV PCR had limited sensitivity for the diagnosis of pneumonia. | - High number of patients and samples studied.<br>- BAL and plasma samples collected consecutively and in parallel, which were prospectively qPCR on fresh samples.<br>- Lung biopsies were included for histological study. | - Viral culture was not used to establish virus replication.<br>- qPCR assay not fully automated (requires prior DNA extraction). |

**Table 1.** *Cont.*

| Article | Material and Methods | Results and Conclusions | Strengths | Limitations |
| --- | --- | --- | --- | --- |
| Iglesias et al., 2017. [15] | - Fifty-six recipients of allogeneic hematopoietic stem cell transplantation (**allo-HSCT, median age: 51 years, range 17–68**) were studied. 16 (28.6%) of these patients presented symptoms of lung disease, and a BAL and plasma sample was collected. BAL was tested by HCMV culture and qPCR, and plasma by qPCR. | - VL of CMV in BAL was positive in 10 (62.5%) patients (in 7 VL >150 copies/mL and in 3 VL <150 copies/mL). <br> - Of the 7 patients with BAL VL >150 copies/mL, 6 (85.7%) were diagnosed with probable HCMV pneumonia, and in 1 (14.3%) case, HCMV disease was discounted. <br> - In 6 patients with HCMV pneumonia (10.7% of all patients), the median BAL VL was 53,250 copies/mL. In 5 cases, the plasma VL was positive, with a median of 538 copies/mL. <br> - Only one BAL sample was positive for CMV by virological culture, with a VL >10,000,000 copies/mL. <br> - All patients' plasma VL was lower than in BAL. <br> - **Any value of CMV VL in BAL**, with compatible signs or symptoms, should be considered suggestive of CMV pneumonia in **allo-HSCT**. | - Fresh BAL samples were used and tested by viral culture. <br> - BAL and plasma samples were collected in parallel, which were prospectively qPCR on fresh samples. <br> - A fully automated and highly sensitive qPCR assay was used. | - Small number of patients with HCMV pneumonitis (6). <br> - Not all patients underwent BAL (only 16). |
| Young Lee et al., 2017. [16] | - In 565 BALs from 565 adult patients (median age: 48 years) with **hematologic malignancies** and signs and/or symptoms of pneumonia who underwent bronchoscopy, HCMV was tested by qPCR, Shell-vial culture and IHC. <br> - Additional assays such as blood HCMV qPCR and lung biopsies were also reviewed. | - There were 464 (82.1%) patients negative for HCMV or <380 copies/mL in BAL. <br> - There were 101 (17.9%) patients with HCVM qPCR >380 copies/mL in BAL. 24 (23.8%) of them were diagnosed with HCMV pneumonia. <br> - Patients with HCMV pneumonia had significantly higher VL in BAL than patients without (7,378,508 vs. 10,899 copies/mL). <br> - Patients with HCMV pneumonia had significantly higher VL in plasma than patients without (683,659 vs. 20,915 copies/mL). <br> - Cut-off value of **28,774 copies/mL** HCMV in BAL was correlated with HCMV pneumonia. | - High number of patients and samples studied. <br> - Fresh BAL samples were used, which were tested by reference techniques such as shell vial culture and immunohistochemical (IHQ). <br> - Lung biopsies were included for histological study. | - Retrospective data analysis. <br> - qPCR assay not fully automated (requires prior DNA extraction). |

**Table 1.** *Cont.*

| Article | Material and Methods | Results and Conclusions | Strengths | Limitations |
|---|---|---|---|---|
| Govender et al., 2017. [17] | - Eighty-seven **infants (median age: 3.7 months)** with suspected HCMV infection and severe pneumonia requiring ventilation, included in the intensive care unit, were tested for HCMV by qPCR on BAL and plasma samples. | - Twenty-nine patients (33.3%) were HCMV-infected and diagnosed with HCMV pneumonitis.<br>- Twenty-five patients (28.7%) were HCMV-infected without HCMV pneumonitis.<br>- Thirty-three patients (37.9%) were HCMV-uninfected.<br>- There was a significant difference in mean HCMV VL in BAL between patients HCMV-infected without pneumonitis (3.78 log10 IU/mL) and those with HMCV pneumonitis (5 log10 IU/mL).<br>- There was a significant difference in mean HCMV VL in plasma between patients HCMV-infected without pneumonitis (3.5 log10 IU/mL) and those with HMCV pneumonitis (4.17 log10 IU/mL).<br>- The threshold of **4.03 log10 IU/mL in BAL** was chosen for predicting HCMV pneumonitis.<br>- qPCR in BAL is more predictive of HCMV pneumonitis than in plasma. | - BAL and plasma samples were collected in parallel, and fresh samples were used for qPCR. | - Viral culture was not used to establish virus replication.<br>- qPCR assay not fully automated (requires prior DNA extraction) and less sensitive than current qPCR. |
| Boeckh et al., 2017. [18] | - Overall, 271 BALs and plasma samples from 271 **hematopoietic stem cell transplant recipients** (132 patients with HCMV pneumonitis and 139 controls) were tested by HCMV qPCR.<br>- All BAL were also tested for the presence of B-globin DNA by PCR. | - Median HCMV VL in BAL of patients with HCMV pneumonitis (2.9 log10 IU/mL) was significantly higher than in the 3 control groups (0 log10 IU/mL in patients with non-CMV pneumonia and idiopathic pneumonia syndrome, and 1.63 log10 IU/mL in asymptomatic patients).<br>- HCMV VL in **BAL > 500 IU/mL** reliably differentiates HCMV disease, good PPV and excellent NPV.<br>- BAL storage time did not appear to affect VL.<br>- Pulmonary haemorrhage, co-pathogens and radiographic presentation do not seem to affect BAL VL. | - High number of cases o HCMV pneumonitis (132).<br>- Fresh BAL samples were used and tested by viral culture.<br>- Analysis of the effect of pulmonary haemorrhage, co-pathogens, and radiological presentation on HCMV VL in BAL.<br>- Analysis of BAL cellularity and quality of DNA extraction by BAL B-globin amplification. | - qPCR performed on BAL samples stored at −80 °C (not fresh).<br>- qPCR assay not fully automated (requires prior DNA extraction).<br>- Plasma samples separated up to 7 days from the performance of the bronchoscopy are included. |

**Table 1.** *Cont.*

| Article | Material and Methods | Results and Conclusions | Strengths | Limitations |
|---|---|---|---|---|
| K. Tan et al., 2015. [10] | - Retrospective analysis of 1074 BALs from 699 **IMS** patients (range 18–92.6 years old), testing HCMV by qPCR and culture (conventional and shell vial).<br>- In 20 cases with positive BAL for HCMV, a lung biopsy was performed for histopathological study. | - Ninety (12.9%) patients were HCMV-positive (PCR and/or culture) in BAL, and 609 (87.1%) were HCMV-negative.<br>- Sensitivity of qPCR (91.3%) was significantly higher than both SV-culture (54.4%) and conventional-culture (28.3%).<br>- Specificity of qPCR (94.6%) was significantly lower than both SV-culture (97.4%) and conventional-culture (96.5%).<br>- NPV of qPCR (99.6%) was significantly higher than both SV-culture (97.9%) and conventional-culture (96.9%).<br>- **VL HCMV in BAL not statistically different between patients with or without pneumonitis.** | - Fresh BAL samples were used and tested by traditional culture and shell-vial.<br>- High number of cases of HCMV pneumonitis were included. | - Retrospective data analysis.<br>- Heterogeneous group of immunocompromised patients included.<br>- qPCR assay not fully automated (requires prior DNA extraction) and less sensitive than current qPCR.<br>- Not all patients underwent HCMV analysis in blood by qPCR.<br>- Not all patients could be studied by histopathology or cytology. |
| Westall et al., 2004. [19] | - Overall, 182 paired samples (BAL and plasma) from 41 **lung transplant recipients (range 18–64 years old)** were tested by HCMV qPCR.<br>- In the BAL samples, qPCR was undertaken using the supernatant and the cell pellet.<br>- Fourty-two patients also underwent a transbronchial biopsy for a histological study of HCMV. | - Fourteen samples (8.1%) had HCMV DNA detected in both BAL and plasma.<br>- Overall, in 123 (71.5%) samples, HCMV DNA was not detected in either BAL or plasma.<br>- In 35 samples (20.3%), HCMV DNA was detected in BAL but not in the plasma.<br>- Qualitative HCMV PCR in BAL low specificity (76%).<br>- Mean VL HCMV in BAL with proven HCMV pneumonitis was significantly higher than those without pneumonitis ($19{,}460 \pm 4917$ vs. $5873 \pm 1543$ copies/mL).<br>- Mean VL HCMV in plasma with proven HCMV pneumonitis was significantly higher than those without pneumonitis ($6791 \pm 2942$ vs. $561 \pm 445$ copies/mL).<br>- All episodes of histologically proven HCMV infection were associated with VL in **BAL > 46,000 copies/mL.**<br>- The detection of HCMV DNA in BAL, compared with detection in plasma, was better correlated with HCMV pneumonitis. | - Patients were enrolled consecutively and prospectively.<br>- BAL and plasma samples were collected in parallel, and fresh samples were used for qPCR.<br>- Lung biopsies were included for histological study. | - Small number of histologically proven HCMV disease (8).<br>- Viral culture was not used to establish virus replication.<br>- qPCR assay not fully automated (requires prior DNA extraction). |

**Table 1.** *Cont.*

| Article | Material and Methods | Results and Conclusions | Strengths | Limitations |
|---|---|---|---|---|
| Chemaly et al., 2004. [20] | - Overall, 43 BALs from 27 **lung transplant recipients** (range 21–65 years old) were tested by shell vial culture and quantitative hybrid capture assay (Q-HCA). <br> - In patients with positive BAL for HCMV, IHC of the transbronchial biopsy was performed. <br> - HCMV was also studied in 18 blood samples using Q-HCA | - 15 (27%) patients had both positive BAL culture and Q-HCA. <br> - 5 (33%) of these 15 were diagnosed as HCMV pneumonitis (positive lung biopsy), all with VL > 500,000 copies/mL in BAL (mean of 1,638,457 copies/mL). <br> - The remaining 10 (66%) without HCMV pneumonitis, all with **VL < 500,000 copies/mL in BAL** (mean of 81,820 copies/mL). <br> - Significantly higher VL in BAL in patients with HCMV pneumonitis compared to those without HCMV pneumonitis (0.002). <br> - Patients with HCMV pneumonitis (5) had a mean VL in the blood of 347,515 copies/mL, while those without pneumonitis (5) had a mean VL in the blood of 3151 copies/mL ($p < 0.02$). <br> - HCMV VL in BAL was more predictive of pneumonitis than the blood VL. | - Patients were enrolled consecutively and prospectively. <br> - Fresh BAL samples were used and tested by traditional culture and shell vial. <br> - All patients underwent lung biopsy for cytological and histological study. | - Small number of patients with HCMV pneumonitis (5). <br> - The quantitative hybrid capture assay (Q-HCA) was used, a less sensitive and automated technique than qPCR. <br> - Not all patients underwent HCMV analysis in blood by qPCR. |
| Chemaly et al., 2003. [21] | - Overall, 42 BALs from 27 **lung transplant recipients** were tested by HCMV quantitative hybrid capture assay (Q-HCA). <br> - In addition, the presence of HCMV in 39 lung biopsies was studied by histology and IHC. <br> - HCMV was also studied in 32 blood samples using Q-HCA. | - HCMV was detected in lung biopsy samples by H&E in 3 (8%) and by IHC in 13 (33%). <br> - VL in BAL was significantly higher than in blood. <br> - HCMV VL in both BAL and blood in patients with histological evidence of HCMV disease was significantly higher than in those without evidence. <br> - Median HCMV VL in BAL was 0 copies/mL in patients with IHQ-negative biopsy, 47,678 copies/mL in biopsy positive-atypical straining and 1,548,827 copies/mL biopsy positive-typical straining ($p < 0.001$). | - Patients were enrolled consecutively and prospectively. <br> - Fresh BAL samples were used. <br> - All patients underwent lung biopsy for cytological and histological study. | - Small number of patients with HCMV pneumonitis. <br> - Viral culture was not used to establish virus replication. <br> - The quantitative hybrid capture assay (Q-HCA) was used, a less sensitive and automated technique than qPCR. <br> - Not all patients underwent HCMV analysis in blood by qPCR. |

**Table 1.** *Cont.*

| Article | Material and Methods | Results and Conclusions | Strengths | Limitations |
|---------|---------------------|------------------------|-----------|-------------|
| Riise et al., 2000. [22] | - Overall, 340 BALs from 35 consecutive **lung transplant recipients (adults and infants)** were tested by HCMV qPCR and culture | - Seventeen (49%) of the patients developed HCMV pneumonitis.<br>- Patients that developed HCMV disease had a significantly higher mean VL in BAL (1120 ± 4.379 copies/mL) compared with those without (180 ± 1177 copies/mL).<br>- **qPCR HCMV in BAL is not useful as a diagnostic tool for HCMV disease.** | - Patients were enrolled consecutively and prospectively.<br>- Fresh BAL samples were used and tested by viral culture. | - Small number of patients with HCMV pneumonitis (17).<br>- qPCR assay not fully automated (requires prior DNA extraction). |

### 4. Conclusions

As shown in this review, in recent years, several studies have been carried out to determine the relationship between the HCMV VL in BAL and the plasma with HCMV pneumonia.

There are some conclusions on which they all agree. PCR techniques were more sensitive and had a higher NPV than culture techniques but were less specific. HCMV DNA detection in BAL does not always imply disease, so quantifying VL should, in theory, be important for clinical interpretation. The mean HCMV load in both BAL and plasma was significantly higher in patients with pneumonitis than in those without. The HCMV load in patients with pneumonitis was higher in BAL than in the plasma, making qPCR in BAL a better predictor of HCMV pneumonitis than in plasma.

Nevertheless, this review highlights the difficulty of establishing a universal VL value, both in BAL and in blood, to differentiate patients with HCMV pneumonia from those without (HCMV DNA shedding). The results and conclusions obtained in these studies are quite different, partly because of the methodological variability of the different studies. In addition, only a few studies used the reference techniques (HCMV detection in biopsy or isolation by culture) to compare viral load results and define HCMV pneumonia cases [16,20]. In each of them, a different value of VL in BAL indicative of active replication in the lung was obtained, ranging from 500 IU/mL [18] to >500,000 copies/mL [20]. On the other hand, some authors concluded that it is not possible to obtain a VL threshold that allows the identification of patients with HCMV pneumonitis [11,22]. Regarding the value of DNAemia testing in the diagnosis of HCMV pneumonia, there are also discrepancies. Some authors found that in up to one-third of patients with pneumonia, HCMV DNA was not detected in the blood [9,19]. Conversely, others claim that in patients in whom HCMV is not detected in plasma, pneumonia can be ruled out [12].

These authors reflect in their work that there are many factors that may influence the VL value in BAL. The first and most important of these is the dilutional effect of bronchoscopy. The serum instilled in the procedure is distributed heterogeneously, and both the volume of lung bathed and the amount of fluid recovered are highly variable among individuals. This makes the quantification in the BAL highly variable. Another important factor is related to the lack of standardization of the tests available in the different laboratories for the quantification of HCMV DNA. In the studies included in the review, it can be observed that different protocols and tests are used in each of them for this purpose. Aspects such as assay performance (limits of detection and quantification), the method for DNA extraction, gene target, and amplicon size contribute to VL variability. In addition, clinically relevant VL values are likely to differ depending on the type of patients and their risk profiles.

Additional caveats include the small number of patients with HCMV pneumonitis in each study. In addition, not all studies used direct diagnostic tests such as viral culture or histology in lung biopsies to establish the level of evidence for CMV pneumonia. Finally, some studies used qPCR assays that were not fully automated and required previous sample handling prior to the extraction of the DNA.

To complete the information available in these studies, prospective multicenter studies would be required. Methodologically, a large number of patients with HCMV pneumonitis would have to be included, and a subclassification of the type of immunosuppression of each patient should be made in order to obtain a VL value in BAL according to the risk of each group. Such a study is, of course, logistically impossible to perform. In addition, in an ideal world, the BAL-related variability would have to be reduced by using a common BAL procedure, another impossible objective. Finally, uniform virologic tests would be needed with all centers participating in the study using the same qPCR assay, which should be as sensitive and automated as possible, and HCMV cultures should be performed on all BAL samples. Of course, concomitant plasma qPCR should be performed, which is the only feasible test currently available in all centers. However, certainly, the most problematic

issue would be the very few cases of histologically proven disease that would be available due to the very low rate of lung biopsies performed, especially in alloHSCT recipients.

**Funding:** This research received no external funding.

**Institutional Review Board Statement:** Not applicable.

**Informed Consent Statement:** Not applicable.

**Data Availability Statement:** Not applicable.

**Conflicts of Interest:** The authors declare no conflict of interest.

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
