# Peer review of "Quantitative PCR for the Diagnosis of HCMV Pneumonia in HSCT Recipients and Other Immunocompromised Hosts"

_hemato, doi:10.3390/hemato4010008_

Round 1
Reviewer 1 Report
The authors summarized the significance of qPCR testing in diagnosing CMV pneumonia in immunocompromised patients such as those after hematopoietic stem cell transplantation, revealing that CMV qPCR, whether using BAL or plasma, has good sensitivity but higher NPV.
1. Patients would benefit greatly from being able to diagnose CMV pneumonia using serum rather than the more aggressive BAL. This point may need to be mentioned.
2. A brief explanation could be given as to the frequency of qPCR testing, e.g., whether it should be done once a week or multiple times a week.
3. Problems of overtreatment (side effects or cost, etc.) for CMV pneumonia based on qPCR diagnosis may be briefly explained.
4. Citations regarding CMV antigenemia versus CMV qPCR testing might be added.
Author Response
- Patients would benefit greatly from being able to diagnose CMV pneumonia using serum rather than the more aggressive BAL. This point may need to be mentioned.
Thank you for your appreciation, but we do not fully agree with it.
For the diagnosis of HCMV pneumonitis it is necessary to detect the virus in lung tissue (proven pneumonitis) or in bronchoalveolar lavage fluid (probable pneumonitis). The presence of HCMV in the blood, together with symptoms and/or signs, is not enough to define proven or probable HCMV disease, with the exception of HCMV retinitis, where ophthalmological findings are very characteristic of CMV chorioretinitis.
In addition, contrary to common opinion, not all patients with HCMV pneumonia have detectable CMV DNA in peripheral blood. Thus, several authors (Leuzinger et al., 2020; Lodding et al., 2017) found that up to one third of patients with HCMV-positive BAL had undetectable plasma loads. Conclusion from these studies was that plasma HCMV qPCR had limited sensitivity for the diagnosis of pneumonia, indicating possible local HCMV replication in the lung. In addition, other authors (Govender et al., 2017; Westall et al., 2004) found that HCMV detection in BAL by qPCR is more predictive of HCMV pneumonitis than plasma qPCR.
In conclusion, qPCR for CMV DNA in peripheral blood can be negative in patients with HCMV pneumonia, and, in addition, there is no quantitative relationship between qPCR in blood and the risk of having HCMV pneumonia. Taken together, peripheral blood PCR nor only does not have the sufficient specificity for the diagnosis of HCMV pneumonia, but it also does not have a sufficient sensitivity.
- A brief explanation could be given as to the frequency of qPCR testing, e.g., whether it should be done once a week or multiple times a week.
Thank you for your comment. This information has been added to the text (lines 122-124).
- Problems of overtreatment (side effects or cost, etc.) for CMV pneumonia based on qPCR diagnosis may be briefly explained.
Thank you for your comment. This information has been added to the text (lines 116-119).
- Citations regarding CMV antigenemia versus CMV qPCR testing might be added.
Thank you for your comment. This information has been added to the text (lines 107-111).
Reviewer 2 Report
The revision proposed by Berengua and Martino is of some interest because it describes and attempts to summarize the diagnosis methodology of HCMV pneumonias and above all to identify the threshold level of HCMV DNA in the BAL. The abstract section is concise, the introduction section is focused on the purpose of the study. The methods are well described. The results are enriched with a summary table of the main works aimed at this topic. The conclusions section is concise. Problems: 1) the title of the work must be modified because it does not correspond to the text reported 2) the authors should report the data as specific to the adult population and remove the reference to the pediatric-only Govender 2017 study. In the case of mixed populations the authors should report the median age. 3) The authors, based on the published results, should differentiate DNAemia obtained from BAL sampling in patients undergoing solid organ transplantation versus those obtained from stem cell transplantation rather than after intensive chemotherapy. In fact, literature data show that patients undergoing solid organ transplantation can "tolerate" in the absence of disease, higher viral loads than patients undergoing stem cell transplantation. On the basis of the literature review, the authors could propose to the scientific community threshold values differentiated according to the type of transplant or the administered immunosuppression. 4) The text should be subjected to a review of the English language.
Author Response
- The title of the work must be modified because it does not correspond to the text reported
We agree with this reviewer since we have not extensively reviewed the diagnosis of HCMV pneumonia but have focused on the caveats of using qPCR. We thus suggest changing the title to:
“Quantitative PCR for the diagnosis of HCMV pneumonia in HSCT recipients and other immunocompromised hosts”
- The authors should report the data as specific to the adult population and remove the reference to the pediatric-only Govender 2017 study. In the case of mixed populations the authors should report the median age.
Thank you for your review. The median age or range information has been added to the table, for those cases where this information was clearly stated in the original article.
- The authors, based on the published results, should differentiate DNAemia obtained from BAL sampling in patients undergoing solid organ transplantation versus those obtained from stem cell transplantation rather than after intensive chemotherapy. In fact, literature data show that patients undergoing solid organ transplantation can "tolerate" in the absence of disease, higher viral loads than patients undergoing stem cell transplantation. On the basis of the literature review, the authors could propose to the scientific community threshold values differentiated according to the type of transplant or the administered immunosuppression
The reviewer raises a very relevant problem. Results of any HCMV-related laboratory test have probably very different interpretations depending on the specific type of immunosuppression the patient has. Thus, studies which focus only in allogeneic HSCT recipients should probably be analysed separately form patients including only SOT recipients. However, we found it impossible to reach to any conclusions trying to differentiate results from different specific scenarios. As an example, the studies by Boeckh et al. and Piñana et al. both focused in alloHSCT. While the former found that a BAL qPCR of 500 IU/mL had a great NPV and a good PPV for the diagnosis of pneumonia, the latter found that the PPV of this same VL in BAL was low, since 60% of patients with a result > 500 IU/mL did not have pneumonia.
- The text should be subjected to a review of the English language.
Thank you for your comment. The English language has been revised.
Reviewer 3 Report
Overall, a well-written review that deals with a very relevant topic in immunocompromised patients. However, the discussion/conclusion section is a little short and could deal in more detail with the question of cutoffs/tresholds or the comparison with biopsy-proven lung tissue. As long as this part is expanded a little, there is no obstacle to publication on my part.
Author Response
Thank you for your feedback. Comments have been added to the discussion.
Round 2
Reviewer 2 Report
The authors fully replied all queries